# Observational Fear Learning in Rats: Role of Trait Anxiety and Ultrasonic Vocalization

**DOI:** 10.3390/brainsci11040423

**Published:** 2021-03-26

**Authors:** Markus Fendt, Claudia Paulina Gonzalez-Guerrero, Evelyn Kahl

**Affiliations:** 1Institute for Pharmacology and Toxicology, Medical Faculty, Otto-von-Guericke University Magdeburg, 39120 Magdeburg, Germany; claudia.gonzalez@st.ovgu.de (C.P.G.-G.); evelyn.kahl@med.ovgu.de (E.K.); 2Center for Behavioral Brain Sciences, Otto-von-Guericke University Magdeburg, 39120 Magdeburg, Germany; 3Integrative Neuroscience Program, Otto-von-Guericke University Magdeburg, 39120 Magdeburg, Germany

**Keywords:** anxiety, observational fear learning, rat, ultrasonic vocalization

## Abstract

Rats can acquire fear by observing conspecifics that express fear in the presence of conditioned fear stimuli. This process is called observational fear learning and is based on the social transmission of the demonstrator rat’s emotion and the induction of an empathy-like or anxiety state in the observer. The aim of the present study was to investigate the role of trait anxiety and ultrasonic vocalization in observational fear learning. Two experiments with male Wistar rats were performed. In the first experiment, trait anxiety was assessed in a light–dark box test before the rats were submitted to the observational fear learning procedure. In the second experiment, ultrasonic vocalization was recorded throughout the whole observational fear learning procedure, and 22 kHz and 50 kHz calls were analyzed. The results of our study show that trait anxiety differently affects direct fear learning and observational fear learning. Direct fear learning was more pronounced with higher trait anxiety, while observational fear learning was the best with a medium-level of trait anxiety. There were no indications in the present study that ultrasonic vocalization, especially emission of 22 kHz calls, but also 50 kHz calls, are critical for observational fear learning.

## 1. Introduction

In potentially threatening situations, rats—as other animals and humans—express a variety of defensive behavior that ultimately helps to survive such situations [1]. The choice of the defensive behavior is dependent on several variables, such as the intensity and the proximity of the threatening stimulus but also on the local conditions, e.g., whether there is a way to escape or a place to hide [2]. A defensive attack may occur if the threat is too close, whereas freezing, a cessation of movements [3], is expressed if the threat is not too close, but escape is not possible. After the initial defensive response, e.g., hiding or freezing, many rats emit ultrasonic vocalizations [1,4,5]. These calls are characterized by a peak frequency around 22 kHz, a narrow bandwidth, and call durations around one second [6,7]. The function of these 22 kHz calls is not completely understood. It is well recognized that they reflect the negative affective state of the emitting rat [8]. Thereby, they serve as species-specific aversive communication signals, i.e., they are recognized by conspecifics and have the capability to change their emotional state [9] or even to warn them about a threat [4]. However, only modest effects are reported in experiments using playbacks of natural 22 kHz calls [10,11,12,13], eventually questioning the direct behavioral effects of 22 kHz calls.

Of note, threatening situations are also associated with very robust and rapid learning processes [14,15]. Both contextual and discrete stimuli that are present during the threatening situation can be associated with the threat, a process called threat or fear-conditioning [16]. Laboratory protocols of fear-conditioning are widely used in neuroscience research to investigate the neural basis of fear/anxiety learning [17,18,19,20]. There is a high translational relevance of this research field since fear-conditioning itself, its specificity, extinction, and inhibition are affected in patients with anxiety disorder [21,22,23]. Some of these changes are not only seen in clinical populations but also in human subjects with high but not yet pathological trait anxiety [24,25,26]. Of note, also rats with high trait anxiety showed exaggerated fear learning, a less specific fear memory, and impaired fear extinction [27,28].

Several studies have described observational fear learning, also called “social fear learning”, “vicarious fear learning”, or “fear-conditioning by proxy”, in rats [29,30,31,32]. In these studies, rats learned fear by observation, i.e., they joined an already fear-conditioned conspecific (the “demonstrator”) in a retention session on conditioned fear and could observe its fear responses to the conditioned context and/or discrete stimulus (that are neutral for the observer rat). To test whether fear was acquired by observation, a further retention test was performed one day later in which the observer rat alone was exposed to the conditioned context or stimulus [33]. Observational fear learning has been successful in male and female cage-mates [30,32], was more pronounced in related and submissive rats [31,32] and less pronounced in dominant rats [31]. Furthermore, the amount of social interaction and ultrasonic vocalization during the observational fear learning session was positively correlated with the observers’ fear response on the following day [30,31,32]. Of note, only 22 kHz calls were analyzed in this experiment [31]. Of potential interest would also be the so-called “50 kHz calls” that are typically emitted during social contacts or other appetitive situations [34,35,36]. Since social interaction was positively correlated with the observers’ fear learning [30,31,32] and 50 kHz calls are typically increased during social contacts, the amount of 50 kHz calls should also positively correlate with the observers’ fear learning.

The role of trait anxiety in observational fear learning has not been investigated in rats so far. There are hints for a possible role of trait anxiety from a human study. In this study, participants had to observe mock panic attacks while their fear responses were measured [37]. “Anxiety sensitivity” was correlated with self-reported fear levels but not physiological arousal during this observational fear challenge procedure. However, potential learning processes during the challenge were not investigated in this study.

In the present study, the role of trait anxiety and ultrasonic vocalization on observational fear learning in male Wistar rats was investigated. Two experiments using a published protocol [33] were performed: In the first experiment, trait anxiety was assessed in the light–dark box test before the rats were submitted to the observational fear learning procedure. In the second experiment, ultrasonic vocalization was recorded throughout the whole observational fear learning procedure and 22 kHz and 50 kHz calls were analyzed. Our working hypotheses were that (1) rats with higher trait anxiety show more pronounced observational fear learning and that (2) higher emission of both 22 kHz and 50 kHz calls in the observational fear learning session leads to enhanced fear memory in the observer rats.

## 2. Materials and Methods

### 2.1. Animals and Housing

Testing was carried out using 102 experimentally naive male Wistar rats aged 8–12 weeks. They were bred and reared at the local animal facility (original breeding stock: Taconic, Silkeborg, Denmark). They were housed in groups of six animals in standard Macrolon Type IV cages (58 cm × 33 cm × 20 cm) with water and standard lab chow (ssniff, Soest, Germany) available ad libitum. Cages were kept in temperature- and humidity-controlled rooms (22 ± 2 °C, 55 ± 10%) with a 12:12 h light/dark cycle (lights on at 6:00 am). All behavioral tests were conducted during the light phase between 8:00 am and 3:00 pm. Our study was carried out in accordance with international guidelines for the use of animals in experiments (2010/63/EU) and was approved by the local ethical committee (Landesverwaltungsamt Sachsen-Anhalt, Az. 42502-2-1587 UniMD).

### 2.2. Apparatus

The light–dark box system consisted of four identical boxes (49.5 cm × 49.5 × 41.5 cm; TSE Systems, Bad Homburg, Germany) divided into two compartments of the same size. The light compartment (135–310 lux) had transparent acryl glass walls, while the dark compartment (0.2–1.5 lux) had black walls. The two compartments were connected by an 8 cm × 6 cm opening. Position and movements of the rats were detected by animal detection infrared sensor frames (detector distance: 14 mm) and analyzed by TSE PhenoMaster software (version 4.9.4).

The fear-conditioning system consisted of four identical transparent acryl glass box (46 cm × 46 cm × 32 cm) located in sound-attenuating chambers (70 cm × 80 cm × 70 cm; TSE Systems, Bad Homburg, Germany). The chambers were equipped with loudspeakers for the acoustic stimuli (background noise of 55 dB SPL and the tone stimuli for fear-conditioning), light sources (continuous illumination of ca. 10 lux), ventilation fans, and video cameras (for monitoring and videotaping) mounted in the ceiling of the chambers. The floor of the boxes consisted of removable stainless steel grids (bars: 4 mm diameter, distance: 9 mm), which were connected to a shock unit and able to deliver foot shocks. Delivery of the different stimuli was controlled by TSE fear-conditioning software (version 09.10). This software also analyzed the position and movements of the rats, recorded by infrared animal detection sensor frames (detector distance: 14 mm). Freezing behavior was defined as no infrared beam crosses for more than 1 s. This automated measurement of freezing was previously validated by demonstrating a high correlation with observer scoring of freezing [13,38].

Recording and analyses of ultrasonic vocalization were performed with the UltraSoundGate system (Avisoft Bioacoustics, Berlin, Germany). For recording, an ultrasound condenser microphone (CM16/CMPA, Avisoft Bioacoustics, Berlin, Germany) sensitive to frequencies of 15–180 kHz (flat frequency response between 25 and 140 kHz; ±6 dB) mounted on one corner of the box was used, which was connected to a laptop via a USB audio device (UltraSoundGate 116H). Acoustic data were recorded by Avisoft Recorder USGH software (version 5.2) using a sampling rate of 250,000 Hz in 16-bit format and a recording range of 0–125 kHz.

### 2.3. Behavioral Procedure

#### 2.3.1. Experiment 1: Role of Trait Anxiety

Forty-eight rats were used in this experiment. On day 1, all rats were tested in the light–dark box (see Figure Figure 1a). Hence, they were placed centrally into the dark compartment and could freely move for 10 min. Then, the rats were returned to their home cages. Based on the results of the light–dark box test, the rats were grouped into sixteen triads, in which one rat served later as the “demonstrator” (DEM), one as the “observer” (OBS), and one as the “naive” (NAIVE). The grouping was done in a way that there were always two triads per cage and that the DEM, OBS and NAIVE rats had similar mean anxiety scores in the light–dark box test (based on time spent in the light compartment).

Three days later, the observational fear learning protocol started. On day 4, the DEM rats were directly fear-conditioned. Therefore, these rats were placed into the fear-conditioning boxes, and after an acclimation time of 10 min, they received three presentations of an auditory stimulus (CS; frequency: 10 kHz, duration: 20 s; mean inter-stimulus interval: 180 s) each co-terminated with an electric stimulus through the floor grids (intensity: 0.7 mA; duration: 0.5 s). Three minutes after the last electric stimulus, the rats were returned to their home cages. One day later (day 5), the observational fear learning session was performed. The DEM rats were returned to the conditioning boxes, each one accompanied by the OBS rat from the same triad. Both rats could freely interact with each other during the following session. After three minutes, the CS was presented three times with a mean inter-stimulus interval of 180 s. There were no electric stimuli during this session. Three minutes after the last CS presentation, the rats were returned to their home cages. Again one day later (day 6), all rats (DEM, OBS, NAIVE) were submitted to a fear retention test. The rats were placed alone into the conditioning boxes and tested with the same protocol as the day before.

#### 2.3.2. Experiment 2: Role of Ultrasonic Vocalization

Fifty-four rats were used in this experiment, i.e., rats were grouped into eighteen triads. The experiment was identical to experiment 1 despite that the rats were not tested in the light–dark box. Furthermore, the box for the observational fear learning procedure was equipped with the ultrasound condenser microphone, and ultrasonic vocalization was recorded throughout the experiment.

### 2.4. Offline Analyses of Behavior and Ultrasonic Vocalization

Automated scoring of behavior via the infrared detection sensor frames could not be used when two rats were together in the box (DEM and OBS in the observational fear learning session). In these cases, the videotapes were used for offline scoring of behavior. The manual scoring was performed by an experienced observer using the Solomon Coder software (version beta 19.08; 16 August 2019; https://solomon.andraspeter.com). Freezing behavior (i.e., cessation of all body movements except those for breathing) of the DEM and OBS rats were scored by two focal sampling sessions per recording.

For the offline analysis of the acoustic data, SASLab Pro software (version 5.2; Avisoft Bioacoustics, Berlin, Germany) was used. After a fast Fourier transformation (512 FFT length, 100% frame, Hamming window and 75% time window overlap), high-resolution spectrograms were produced with a frequency resolution of 488 Hz and a time resolution of 0.512 ms. Onset and offset of the recorded 22 kHz calls were manually marked and 50 kHz calls (all calls > 30 kHz) were counted and categorized by a person, who was not aware of the experimental condition. The following parameters were determined and calculated for each single session: latency of the first 22 kHz call from start of the session, number of 22 kHz calls, mean 22 kHz call duration, total 22 kHz call duration/session, mean peak frequency of 22 kHz calls, and number of 50 kHz calls (peak frequency greater than 30 kHz). Furthermore, the call profiles were determined by visually sub-categorizing the 50 kHz calls into the following categories originally described by Wright and colleagues [39]: (i) downward ramp: monotonically decreasing frequency; (ii) upward ramp: monotonically increasing frequency; (iii) flat: near-constant frequency; (iv) complex: two or more directional changes in frequency; (v) trills: oscillation with a period of ca. 15 ms, can be flanked by a monotonic portion or can contain higher-frequency components; (vi) short: duration less than 12 ms; (vii) steps: one or more instantaneous frequency change to a higher or lower frequency; (viii) inverted-U: monotonic frequency increase followed by a monotonic decrease; (ix) composite: calls that compromise more than one category.

### 2.5. Descriptive and Analytical Statistics

Behavioral data are expressed as means ± standard errors of the mean (SEM), whereas acoustic data are shown as whisker box plots. Statistical analyses were performed with GraphPad Prism (version 8.00, GraphPad Software Inc., La Jolla, CA, USA). Data were checked for normal distribution (D’Agostino and Pearson’s omnibus normality test) and were analyzed by analysis of variance (ANOVA) or Kruskal–Wallis test followed by appropriate post hoc comparisons (Holm–Sidak’s or Dunn’s multiple comparison tests) or Mann–Whitney tests. A *p* < 0.05 was considered statistically significant.

## 3. Results

### 3.1. Experiment 1: Role of Trait Anxiety

In this experiment, the rats were first submitted to a light–dark box test (Figure 1a). Then, the rats were grouped with respect to their prospective roles (DEM, OBS, NAIVE; *n* = 16/group) in the upcoming observational fear learning experiment with the restriction that there must be two triads with DEM OBS and NAIVE rats per cage. Figure 1b shows the mean percent time the rats of the different groups spent in the light compartment of the light–dark box (lower values indicate higher anxiety). An ANOVA revealed no differences between the groups (*F*_2,45_ = 0.06, *p* = 0.95) and a Bartlett’s test no differences in variances (*χ*^2^ = 2.78, *p* = 0.25).

As expected, there were strong differences in the time spent freezing during the retention test on conditioned fear (Figure 1c; ANOVA: *F*_2,45_ = 50.08, *p* < 0.0001). Post hoc comparisons revealed higher freezing scores in the DEM rats compared to the OBS and NAIVE rats (Holm–Sidak’s tests: *t*_16_ = 7.12, *p* < 0.0001 and *t*_16_ = 9.65, *p* < 0.0001, respectively). Importantly, the OBS rats had higher freezing levels than the NAIVE rats (*t*_16_ = 2.54; *p* = 0.015), indicating observational fear learning in the OBS rats.

Next, we were interested if anxiety-like behavior in the light–dark box was associated with the freezing behavior in the retention test on conditioned fear. Figure 1d depicts the freezing behavior of the individual rats as a function of the percent time the rats spent in the light compartment of the light–dark box. We tested several models for linear and nonlinear regressions and found best-fits for a simple linear regression in the DEM group (R^2^ = 0.52, *p* = 0.002) and a Gaussian curve OBS group (R^2^ = 0.58). For the NAIVE group, we did not find a model with R^2^ > 0.20. This indicates that in DEM rats conditioned fear is higher with higher anxiety scores (i.e., less time spent in the light compartment), while in OBS rats conditioned fear is highest in medium-anxious rats. This is supported by further analyses in which we sorted the rats within the DEM, OBS and NAIVE groups regarding their anxiety scores in the light–dark box test. The thirds of rats with most percent time spent in the light compartment formed the low anxiety subgroups, the thirds of rats with the lowest time spent in the light compartment formed the high anxiety subgroups, and the remaining rats formed the medium anxiety subgroups (Figure 1e–g). Separated ANOVAs for DEM, OBS, and NAIVE rats revealed effects of trait anxiety in DEM (*F*_2,13_ = 4.04, *p* = 0.04) and OBS (*F*_2,13_ = 12.87, *p* = 0.0008) rats, but not in NAIVE rats (*F*_2,13_ = 0.04, *p* = 0.96). Post hoc comparisons showed significantly more freezing in high-anxiety DEM rats compared with low-anxiety DEM rats (Figure 1e; *t*_9_ = 2.83, *p* = 0.04), and more freezing in medium anxious OBS rats compared to low- and high-anxiety OBS rats (Figure 1f; *t*_9_ = 3.61; *p* = 0.006 and *t*_9_ = 4.12, *p* = 0.001, respectively).

Last, to evaluate whether the differences in trait anxiety levels between the DEM and OBS rats affect observational fear learning, we also calculated the differences between the anxiety scores (% time spent in the light compartment) of the DEM-OBS pairs that had the observational fear learning session together. Using these differences, we built DEM-OBS subgroups, in which the DEM rat was clearly more anxious than the OBS rat (“DEM more anxious”, difference > 10%), in which DEM and OBS rats were similarly anxious (“none more anxious”, difference < 10%), and in which the OBS rat was more anxious than the DEM rat (“OBS more anxious”, difference > 10%). As indicated in Figure 1h, the trait anxiety difference within the DEM-OBS pairs did not affect the freezing of the OBS rat in the retention test (ANOVA: *F*_2,13_ = 0.58; *p* = 0.57).

### 3.2. Experiment 2: Role of Ultrasonic Vocalization

In this experiment, the rats (*n* = 18/group) were not tested in the light–dark box. The observation fear learning protocol was identical to the one used in experiment 1. However, ultrasonic vocalization was recorded throughout the experiment (Figure 2a). As expected and already observed in experiment 1, there were significant group differences in the fear retention test (Figure 2b; ANOVA: *F*_2,51_ = 100.90; *p* < 0.0001). DEM rats expressed more freezing behavior than OBS and NAIVE rats (Holm–Sidak’s tests: *t*_18_ = 11.03, *p* < 0.0001 and *t*_18_ = 13.27, *p* < 0.0001, respectively). Of note, OBS rats had significantly higher freezing scores than NAIVE rats (*t*_18_ = 2.24, *p* = 0.03), indicating successful observational fear learning in the OBS rats.

During all phases of this experiment, ultrasonic vocalization was recorded and analyzed. Our first analysis was focused on 22 kHz calls (Figure 3 and Figure 4). 22 kHz calls were exclusively emitted by DEM rats with the exception of one NAIVE rat (four 22 kHz calls in the retention test). The number of DEM rats emitting 22 kHz calls in the different phases of the experiment was not different (Figure 3a; chi-squared test: *χ*^2^ = 1.99, *p* = 0.37). From the fourteen DEM rats that emitted at least once during the experiment 22 kHz calls, seven rats emitted them in all three phases. The other seven rats emitted 22 kHz calls mainly in the observational fear learning session (75%), i.e., when a conspecific was present, and less during the sessions without conspecific (50%). It should be emphasized that the recordings from the observational fear learning phase potentially include vocalizations from two rats, the DEM and the OBS rats. However, since we never detected simultaneously emitted 22 kHz calls in the sonograms and the OBS rats never emitted 22 kHz calls in the retention test, we are confident that all recorded 22 kHz calls in the observational fear learning phase were emitted by the DEM rats.

The latency of the first 22 kHz call was very different in the three sessions (Figure 3b). Latencies were longest during the fear-conditioning session. The 22 kHz calls were never emitted before the first electric stimulus but usually appeared first after the 2nd electric stimulus (in fewer cases also after the 1st or third stimulus). The observational fear learning session and the retention test had identical protocols (including no electric stimuli); therefore, these sessions can be directly compared. The latencies of the first 22 kHz calls in the observational fear learning session were significantly longer than those in the retention tests (Mann–Whitney test: *U* = 29, *p* = 0.03). In the observational fear learning session, most rats started to emit 22 kHz calls after the first tone presentation, while in the retention test, most rats started before or during this tone.

Despite the different latencies, the number of 22 kHz calls during the different phases of the experiments was not different (Figure 3c; Kruskal–Wallis test: *H* = 1.00, *p* = 0.61). When we compared the number of 22 kHz calls in the observational fear learning session with the freezing behavior of DEM or OBS rats in the retention tests (Figure 3d), we did not find any correlations with acceptable R^2^-values (for example, linear regression: both R^2^ = 0.03). However, after grouping the observational fear learning sessions-in-sessions with low (0–1), medium (21–121), and high number (215–590) of 22 kHz calls (lower, medium and higher third, i.e., *n* = 6/group), we found that the number of 22 kHz calls in the observational fear learning sessions affected the freezing behavior of DEM rats in the retention test (Figure 3e; ANOVA: F_2,15_ = 4.61, *p* = 0.03). Post hoc comparisons revealed that DEM rats from observational fear learning sessions with medium or high numbers of 22 kHz calls showed higher freezing scores in the fear retention test than those from a session with low numbers (Holm–Sidak’s tests: *t* values > 2.58, *p* values = 0.05). Against that, the freezing of OBS rats in the retention test was not affected by the number of 22 kHz calls in the observational fear learning sessions (Figure 3f; ANOVA: *F*_2,15_ = 0.26, *p* = 0.77).

We further analyzed several parameters of the 22 kHz calls (total number: 6725 calls). The mean call duration was approximately 0.90–1.00 s (Figure 4a), and the peak frequency of the calls approximately 21.8–22.2 kHz (Figure 4b). These parameters were not affected by the phase of the experiment (ANOVAs: *F*_2,30_ = 0.10, *p* = 0.91 and *F*_2,30_ = 0.31, *p* = 0.73, respectively). Regularly, calls with very short durations were recorded (2.3% of the calls were <300 ms and 0.8% < 150 ms; Figure 4c–e), and about 0.2% of the calls had an exceptionally long duration (>3000 ms). The maximal call duration in our experiment was 4.341 s (Figure 4f).

In contrast to the 22 kHz calls, the 50 kHz calls were emitted by all rats (DEM, OBS, NAIVE) in all phases of the experiment (Figure 5a,b shows some representative examples of 50 kHz calls). Clearly, the highest number of 50 kHz calls were emitted in the observational fear learning phase. Of note, two rats were present during this phase, and very likely, both of them emitted 50 kHz calls. A comparison of this phase with the other phases, in which always only one rat was present, is, therefore, difficult, if not impossible. However, such an analysis is possible for the retention test, and this analysis revealed different numbers of 50 kHz calls in DEM, OBS, and NAIVE rats (Kruskal–Wallis test: *H* = 14.21 *p* = 0.0008). Pairwise comparisons show less calls in DEM and OBS rats than in NAIVE rats (*p* = 0.002 and *p* = 0.02, respectively), but no difference between DEM and OBS rats (*p* = 0.12). Figure 5b depicts the 50 kHz call category profile (Figure 5b) using the categories described by Wright and colleagues [39], see examples in Figure 5c. The call profile significantly changed across the different sessions of the experiment. Of note, downward and upward ramp calls were emitted less often in the observation fear learning session and in the retention test of the DEM rats than in the other phases (*χ*^2^ tests: *χ*^2^ values > 67; *p* values < 0.0001). In contrast, the proportion of complex calls and trills were higher in these two phases (*χ*^2^ test: *χ*^2^ values > 125; *p* values < 0.0001).

We further analyzed whether the number of 50 kHz calls in the observational fear learning session was correlated with the freezing of DEM or OBS rats in the retention test (Figure 5d). Again, no correlations with acceptable R^2^-values were found (for example,: linear regression: R^2^ = 0.04 or 0.02, respectively). In addition, grouping the observational fear learning sessions-in-sessions with low (7–34), medium (51–113), and high number (137–1047) of 50 kHz calls (*n* = 6/group) did not reveal effects of 50 kHz calls on the freezing behavior of DEM rats (Figure 5e; ANOVA: *F*_2,15_ = 0.07, *p* = 0.93) or OBS rats (Figure 5f; ANOVA: *F*_2,15_ = 0.77, *p* = 0.48) in the fear retention session.

## 4. Discussion

The present study investigated the role of trait anxiety and ultrasonic vocalization on observational fear learning in rats. For our experiments, we used a published protocol [33] in which rats from the same cage were grouped in triads (DEM, OBS, NAIVE). The DEM rat was directly fear-conditioned, the OBS rat had to learn fear by observing the DEM rat, and the NAIVE rats stayed naive until the fear retention test. In both retention tests of the present study, DEM rats expressed a freezing behavior, followed by the OBS rats and the NAIVE rats (Figure 1c and Figure 2b). Importantly, OBS rats had significantly higher freezing scores than the NAIVE rats indicating successful observational fear learning in the present experiments. Very similar differences were observed when freezing behavior during context exposure only or during the tone presentations was separately analyzed (data not shown), supporting previous data showing that both cued and contextual fear can be learned by observation [32]. It is important to note that in the used observational fear learning protocol, the OBS rats learned by observing a learned fear response that the DEM rats expressed during the observational fear learning session in which no unconditioned stimuli (the electric stimuli) are presented. This is different in other protocols in which the OBS rats observed the DEM rats directly during fear-conditioning, i.e., during the application of aversive stimuli that usually induce strong unconditioned responses [40,41] or in which observing fear responses affects later fear-conditioning in the OBS rats [42]. Importantly, all of these protocols are based on the concept that the recognition of the DEM rats’ fear triggers a similar feeling in the OBS rats, i.e., an empathy-like or anxiety state, which then leads to a learning experience or influences a following learning process [43,44]. However, the different protocols may lead to different subtypes of observational learning, and trait anxiety and ultrasonic vocalization may play different roles in these different learning subtypes.

Our first experiment was focused on the role of trait anxiety. The present data show that trait anxiety differently affected the fear response of DEM and OBS rats. In DEM rats, we observed a linear correlation of trait anxiety and fear learning that was also obvious after subgrouping the DEM rats in low, medium and high anxiety. The more anxious the subgroup was the more freezing in the retention test they expressed (Figure 1d,e). This finding is in line with previous publications showing that trait anxiety in laboratory rodents is positively correlated with the strength of fear learning [27] and that breeding lines or strains with higher trait anxiety show enhanced fear expression [28,45,46]. For the OBS rats, we expected a similar relationship between trait anxiety and observational fear learning. However, OBS rats with a medium level of trait anxiety expressed highest freezing responses in the retention test, while low- and high-anxiety OBS rats expressed very low freezing responses (Figure 1d,f). Of note, mean trait anxiety was not different in DEM, OBS, and NAIVE rats (Figure 1b) and played no role in the freezing behavior of NAIVE rats (Figure 1g).

Our study strongly indicates that observational fear learning is best in medium anxious OBS rats and relatively poor or absent in low- and high-anxiety OBS rats. To the best of our knowledge, the present study is the first one investigating the role of trait anxiety in observational fear learning. We also did not find such studies in other species, including humans or on other observational learning processes. Actually, we only find one human study showing that trait anxiety of parents, who served as demonstrators in this study, was positively correlated with observational fear learning of the parents’ kids [47]. This effect of demonstrators’ trait anxiety on observers’ observational fear learning was not seen in the present study after grouping the DEM rats into low, medium and high-anxiety subgroups (data not shown).

One explanation of the present data could be that both too low and too high anxiety levels interfere with several processes that are important for observational fear learning. These processes include recognizing the DEM rats’ emotional state, experiencing a similar emotional state (an empathy-like or anxiety process), associating this state with the stimuli that triggered the DEM rats’ state, and consolidating such associations. For example, OBS rats with too low trait anxiety might simply not develop this empathy-like or anxiety process, which is necessary to learn by observation. In contrast, OBS rats with high trait anxiety might develop too much fear/anxiety during the observational fear learning session, which then interferes with learning abilities and/or lead to very unspecific learning. The role of trait anxiety in these processes is poorly investigated in laboratory rodents and even in humans poorly understood. For example, one study shows that high-anxiety participants recognize fear faces better than low anxious participants [48], while another study did not find this effect of trait anxiety [49]. Unfortunately, very often, only low- and high-anxiety participants are compared in these studies, and a medium level of anxiety is neglected.

We also had the suspicion that the relationship between the DEM rats’ and OBS rats’ trait anxiety plays a role in observational fear learning, i.e., that OBS rats maybe learn better from DEM rats that are more or less anxious than they are or that similar levels of trait anxiety in DEM and OBS rates are optimal. However, we did not find that any of these relationships lead to better observational fear learning (Figure 1h), even though it may be beneficial for observational fear learning if the demonstrator is more anxious than the observer.

The aim of our second experiment was to get some insight into the role of ultrasonic vocalization during observational fear learning. Our hypothesis was that both 22 kHz calls and 50 kHz calls should play a role in observational fear learning. The 22 kHz calls should be of impact since they are discussed as species-specific aversive communication signals [4,9,50], while 50 kHz calls should matter since they are typically emitted during positive social interaction [34,35,36], and the amount of social interaction was found to be correlated with observational fear learning [30,31,32]. Actually, Jones and colleagues showed a correlation of the total duration of 22 kHz calls in the observational fear learning session with the OBS rats’ freezing during the retention test [31]. As in the aforementioned study, we recorded 22 kHz calls only in DEM rats—with the exception of one NAIVE rat emitting four 22 kHz calls during the retention test. Of note, 22 kHz calls during the observational fear learning session could potentially also be emitted by the OBS rats. However, analyses of the video recordings of this session revealed that there were always the DEM rats and rarely the OBS rats showing the typical body posture and flank movements that are associated with the emission of 22 kHz calls. Furthermore, no simultaneously emitted 22 kHz calls were identified in the sonograms. Therefore, we are confident that all recorded 22 kHz calls in the observational fear learning session were emitted by the DEM rats.

During the fear-conditioning session and the retention test, about half of the DEM rats emitted 22 kHz calls, while more of them (72%) emitted these calls during the observational fear learning session, i.e., when a conspecific was present (Figure 3a). This difference was not statistically confirmed but could indicate a so-called “audience effect”. The audience effect refers to the idea that 22 kHz calls serve as alarm calls to conspecifics and, therefore, require their presence [4,9]. While the audience effect was described in one of the seminal studies on rats’ ultrasonic vocalization [4], this effect was not observed in other studies [51]. In addition to the potential audience effect, we found that the latency to emit 22 kHz calls was longer in the observational fear learning session than in the retention test (Figure 3b). This effect was most probably caused by the social interaction with the OBS rats and could, therefore, reflect social buffering of fear [52,53]. However, the number of 22 kHz calls was not different in the different phases of the experiment (Figure 3c). Of note, there was also no association of 22 kHz call emission during the observational fear learning session with the OBS rats’ fear in the retention test (Figure 3d,f), which stands in contrast to a previous report that found such a correlation [31]. However, we found that DEM rats that emitted medium and high number of 22 kHz calls in the observational fear learning session express more fear in the retention test than the DEM rats from the session with a low number of 22 kHz calls (Figure 3e). Very similar results were obtained when the total duration of 22 kHz calls was analyzed (data not shown).

The analyses of the call parameters of the recorded 22 kHz calls showed no differences to the 22 kHz calls described in the literature [6,7,54], i.e., the mean duration was between 0.87 and 0.99 s and the peak frequency between 21.8 and 22.2 kHz calls in the different phases of the experiment. These parameters were not affected by the phase of the experiment. As already previously reported [6], we observed a high variation in the duration of the 22 kHz calls. Approximately 2.3% of the analyzed 22 kHz calls were shorter than 300 ms, and about a third of them (0.8%) were even shorter than 150 ms, two thresholds that are sometimes used [6,9]. In our opinion, these thresholds are arbitrary and not justified since such short calls occurred regularly within sequences of longer calls, without any “external disturbances”. However, some of the short 22 kHz calls were obviously caused by such external disturbances, such as interactions with the OBS rats or the onset and offset of the tone stimulus or the electric stimulus. Rarely (0.2%), calls with call durations longer than 3 s were recorded. Of note, the maximal duration in our study was 4.341 s (Figure 4f), which is ca. 400 ms longer than the published maximal call duration [6].

In contrast to the 22 kHz calls, 50 kHz calls were emitted by DEM, OBS, and NAIVE rats and recorded in all phases of the experiment. The number of calls varied strongly in the different phases (Figure 5a). Most calls were recorded in the observational fear learning session, which is not surprising since two rats were present in this phase. Of note, we cannot distinguish which rat emitted the 50 kHz calls in this session. Only a little amount of 50 kHz calls were emitted in the fear-conditioning session. Usually, these calls were present until the first aversive electric stimulus, then no calls were emitted for a while and then, often 22 kHz calls were emitted (see above). However, 50 kHz calls were also occasionally recorded a few seconds after the electric stimuli. In the retention test, the number of 50 kHz calls was similar in DEM and OBS rats, while NAIVE rats emitted higher amounts of 50 kHz calls. Since 50 kHz calls are associated with positive emotional states [34,36,55,56], this difference between OBS and NAIVE rats may indicate observational fear learning in the OBS rats. However, 50 kHz calls are also emitted during exploratory behavior [35]; the high numbers of 50 kHz calls in NAIVE rats could also be explained by the fact that the retention test was the first exposure of the naive rats to the experimental setup.

Based on the idea that the number of 50 kHz calls during the observational fear learning session may indicate the amount of social interaction between DEM and OBS rats, we also analyzed whether the number of 50 kHz calls in this session is correlated with the freezing behavior in the retention tests. This was neither the case for DEM nor for OBS rats (Figure 5d–f). We also analyzed the 50 kHz call profiles in the different sessions using the categories published by Wright and colleagues [39], Figure 5c. Actually, the call profile significantly changed across the different sessions (Figure 5b). Of note, the profiles of the observational fear learning session and of the retention test of the DEM rats were similar by consisting of more complex calls and trills and fewer downward and upward ramp calls than the other sessions. So far, the meaning of the different 50 kHz call categories is poorly understood. Some categories seem to be associated with certain behaviors, e.g., split and composite calls with running and jumping and trills with slower movements [57] but also with playful attacks in juvenile rats [58]. We detected trills and complex calls more in the observational fear learning session, i.e., in a social situation, but surprisingly also in the retention test of the demonstrators. The presence of these call types in the latter session maybe indicate the search for the social partner the DEM rats had in the session before. Upward and downward ramp calls were most emitted in the fear-conditioning session (before the aversive electric stimuli) and during the retention tests of OBS and NAIVE rats. During these sessions, mainly exploratory behavior was expressed, which suggests that these two call types might be associated with exploration.

Together, these findings suggest no important roles of 22 kHz calls and 50 kHz calls in observational fear learning. We think that the number, total duration and latency of 22 kHz calls potentially signal the strength of conditioned fear of the DEM rats in the different phases of the experiment. Furthermore, there were minor hints for audience effects as well as for social buffering with respect to 22 kHz calls in the observational fear learning phase. In this phase, 50 kHz calls were emitted in a high amount, but this is expected in a situation in which two rats can socially interact. The lower number of 50 kHz calls in DEM and OBS rats in the retention test may indicate a less appetitive state compared with the naive rats and thereby support that OBS rats learned fear by observation.

Based on the present findings, the question arises: Which information source is used by the OBS rats during observational fear learning if 22 kHz calls and 50 kHz calls are not critically involved? The DEM rats express several species-specific behavioral signs of fear, including freezing (in mice, even facial expression of fear was described [59]), which may serve as visual stimuli for observational fear learning [43]. Of note, rats also release an alarm pheromone during aversive emotional states that can induce defensive behavior in conspecifics [60,61,62]. During social interaction, this alarm pheromone might be best perceived by the OBS rats, which could be the reason why social interaction during the observational fear learning session correlates with the learned fear in OBS rats [30,31]. Ultimately, these visual and/or olfactory stimuli are most likely sufficient for the OBS rats to learn fear by observation.

## 5. Conclusions

The present study showed that trait anxiety affects direct fear learning and observational fear learning slightly differently. Direct fear learning was more pronounced with higher trait anxiety (DEM rats), while observational fear learning was the best with a medium-level of trait anxiety (OBS rats). There were no indications in the present study that ultrasonic vocalization, especially emission of 22 kHz calls, is critical for observational fear learning. We believe that both 22 kHz calls and 50 kHz calls are indicative of the rats’ individual emotional state but play no major role in observational fear learning.

## Figures and Tables

**Figure 1 brainsci-11-00423-f001:**
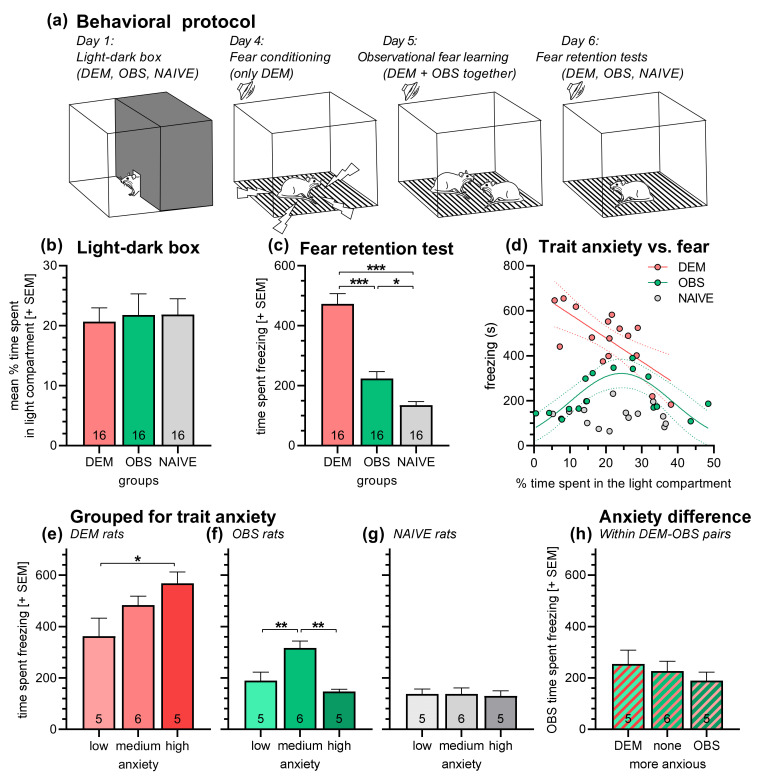
Role of trait anxiety in observational fear learning. (**a**) Behavioral protocol (for more details, see Materials and Methods): After a light–dark box test, rats were divided into three groups (“demonstrator” (DEM), “observer” (OBS), “naive” (NAIVE)) with similar anxiety levels. DEM rats were fear-conditioned. One day later, the DEM and OBS pairs were exposed to the conditioning boxes. Last, all rats were individually tested for conditioned fear. (**b**) Mean percent time spent in the light compartment of the light–dark box. DEM, OBS, and NAIVE rats had similar anxiety levels. (**c**) DEM rats expressed more freezing in the retention test than rats of the two other groups. Importantly, OBS rats had significantly higher freezing scores than NAIVE rats, indicating observational fear learning in OBS rats. (**d**) Freezing scores as a function of trait anxiety (more% time spent in the light compartment indicate less anxiety). In DEM rats, higher freezing scores could be observed with higher trait anxiety. In OBS rats, a bell-shaped correlation was found, freezing was highest with a medium levels of trait anxiety. These correlations were confirmed by grouping the animals into low, medium, and high anxiety. (**e**) High anxious DEM rats had significantly higher freezing scores than low-anxiety DEM rats. (**f**) Medium anxious OBS rats had significantly higher freezing scores than low- and high-anxiety OBS rats. (**g**) Trait anxiety did not affect freezing scores in NAIVE rats. (**h**) The difference of trait anxiety within a DEM-OBS pair did not influence freezing scores in the retention test. In this analysis, the DEM-OBS pairs were separated into subgroups in which DEM rats being more anxious than the OBS rat (“DEM more anxious”), DEM and OBS rats that were similarly anxious (“none more anxious”), and OBS rats that were more anxious than the DEM rats (“OBS more anxious”). Abbreviations: DEM, demonstrator rats; OBS, observer rats; NAIVE, naive rats. Numbers in the bars indicate group and subgroup sizes. *** *p* < 0.001, ** *p* < 0.01, * *p* < 0.05; comparisons as indicated, after significant main effects in ANOVA.

**Figure 2 brainsci-11-00423-f002:**
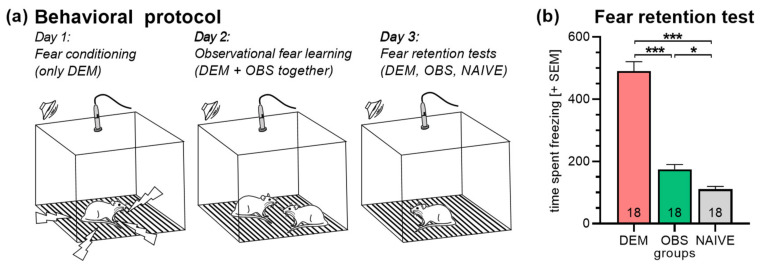
(**a**) Behavioral protocol of experiment 2 was identical to experiment 1, but no light–dark test was performed, and ultrasonic vocalization was recorded throughout the experiment. (**b**) In the retention test, DEM rats showed more freezing than OBS and NAIVE rats. OBS rats had significantly higher freezing scores than NAIVE rats, indicating observational fear learning in OBS rats. Abbreviations: DEM, demonstrator rats; OBS, observer rats; NAIVE, naive rats. Numbers in the bars indicate group sizes. *** *p* < 0.001, * *p* < 0.05; comparisons as indicated, after significant main effects in ANOVA.

**Figure 3 brainsci-11-00423-f003:**
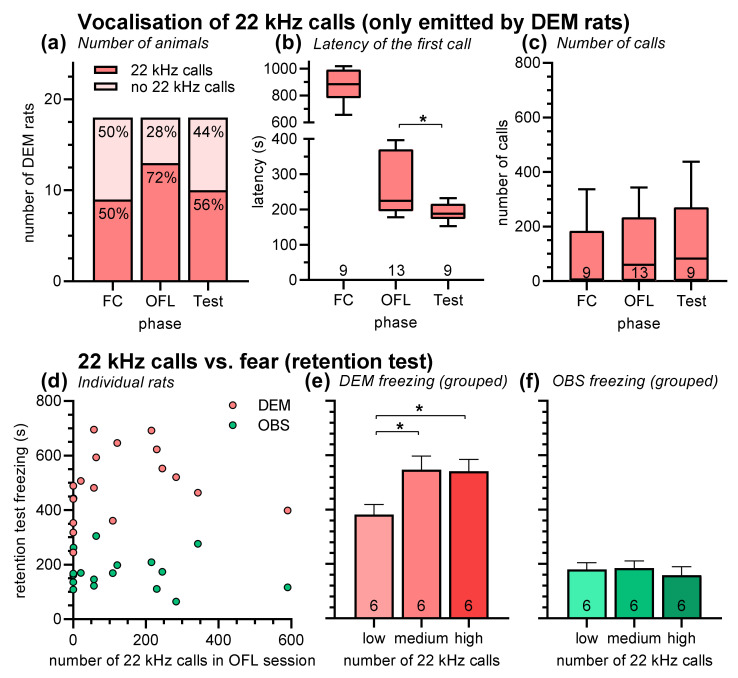
(**a**) Number of rats emitting 22 kHz calls, (**b**) latency of the first 22 kHz call, and (**c**) number of 22 kHz calls in the different phases of the experiment. 22 kHz calls were almost exclusively emitted by DEM rats. There was no difference in the number of rats emitting calls and in the number of calls. However, latency was longest in the fear-conditioning session and significantly longer during the observational fear learning session than during the retention tests. (**d**) Freezing behavior of DEM and OBS rats in the retention session as a function of the number of 22 kHz calls in the observational fear learning session. No correlations were found. (**e**) Next, the observational fear learning session were grouped into sessions with low, medium, and high number of 22 kHz calls. DEM rats of OFL sessions with low number of 22 kHz calls showed lower freezing behavior than the DEM rats from the other session. (**f**) Freezing behavior of OBS rats was not affected by the number of 22 kHz calls in the observational fear learning session. *Y*-axis scale and units in panels (**e**,**f**) are the same as in panel (**d**). Abbreviations: DEM, demonstrator rats; FC, fear-conditioning session; OBS, observer rats; OFL, observational fear learning session; Test, retention test. Numbers in or below the bars or boxes indicate group and subgroup sizes. * *p* < 0.05; comparisons as indicated, after significant main effects in Kruskal–Wallis test or ANOVA, respectively.

**Figure 4 brainsci-11-00423-f004:**
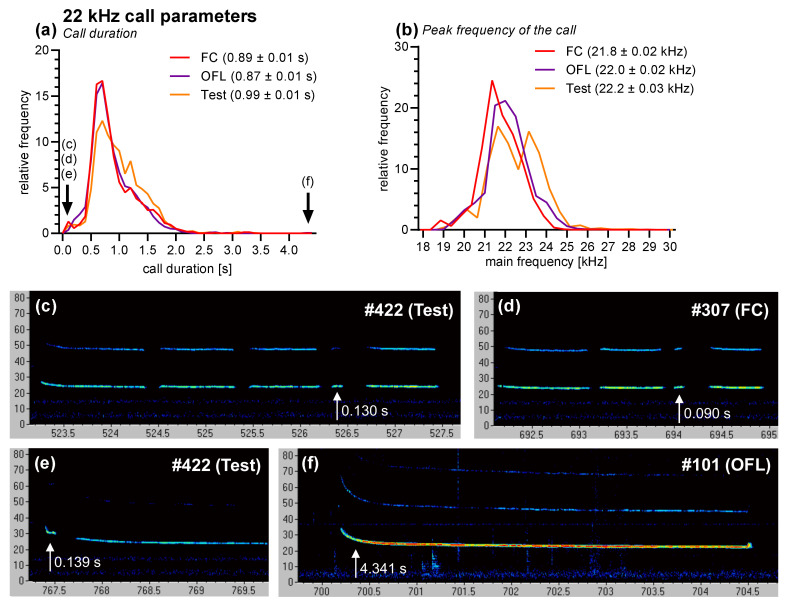
Parameters of the 22 kHz calls. Histogram depicting (**a**) the distribution of the call duration and (**b**) peak frequencies of the calls in the different phases (mean ± SEM). (**c**–**f**) Examples of calls with very short (**c**–**e**) or exceptionally long (**f**) call duration. Abbreviations: FC, fear-conditioning session; OFL, observational fear learning session; Test, retention test.

**Figure 5 brainsci-11-00423-f005:**
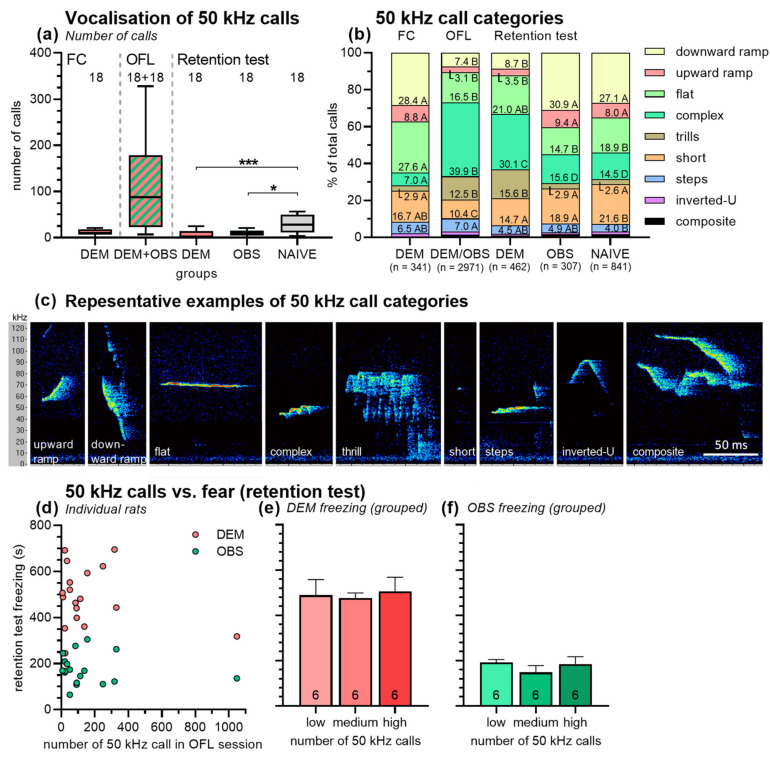
(**a**) Number of 50 kHz calls in the different phases of the experiment. Most 50 kHz calls were emitted in the observational fear learning session in which two rats were present. In the retention test, NAIVE rats emitted more 50 kHz calls than DEM and OBS rats. (**b**) 50 kHz call category profiles in the different sessions. The different colors of the bar sections indicate the different 50 kHz call categories. The numbers in the sectors represent the percent number of calls in a given category, related to the total number of calls in the respective session (indicated on the bottom). The letters after the number indicate whether the proportion of a call category differs across the sessions (*χ*^2^ test). Different letters symbolize significantly different proportions; equal letters stand for non-significant proportions. (**c**) Representative examples of the different 50 kHz call categories. (**d**) Freezing behavior of DEM and OBS rats in the retention session as a function of the number of 50 kHz calls in the observational fear learning session. No correlations were found. (**e**,**f**) Therefore, the observational fear learning sessions were grouped into sessions with low, medium, and high number of 50 kHz calls. No effects of 50 kHz calls on the freezing behavior of DEM and OBS rats in the retention test were found. *Y*-axis scale and units in panels (**e**,**f**) are the same as in panel (**d**). Abbreviations: DEM, demonstrator rats; FC, fear conditioning session; OBS, observer rats; OFL, observational fear learning session; NAIVE, naive rats. Numbers in the bars or above the boxes indicate group and subgroup sizes. *** *p* < 0.001, * *p* < 0.05; comparisons as indicated, after significant main effects in Kruskal–Wallis test.

## Data Availability

The data presented in this study are available on request from the corresponding author.

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
