# Peer review of "Observational Fear Learning in Rats: Role of Trait Anxiety and Ultrasonic Vocalization"

_brainsci, 2021, doi:10.3390/brainsci11040423_

Round 1

Reviewer 1 Report

The manuscript by Drs Fendt and associates reported 1) the effect of trait anxiety on direct fear learning and observational fear learning and 2) the role of two types of communicative ultrasonic vocalizations (22-kHz calls and 50-kHz calls) in observational learning in laboratory rats. They showed that 1) the level of trait anxiety had a positive correlation with direct fear learning, in contrast, a middle level of trait anxiety was the most suitable for observational fear learning and 2) observational fear learning was not significantly related to the degree of exposure to ultrasonic vocalizations during learning sessions. Their findings suggest that 1) direct fear learning and observational fear learning are differently influenced by levels of trait anxiety and 2) ultrasonic vocalizations possess little communicative value in observational fear learning.

The design of this study as well as most of the results are clear and presented in a largely well-written manner. Comments to the authors are as follows.

  1. I think that it is proper to add one more observer group, each animal in which is accompanied during observational learning session by an unpaired demonstrator rat received CS (auditory and contextual stimuli) and foot shock separately, because, strictly speaking, it is unclear in the current study whether observational fear learning could be attributable to negative affective states by means of fear conditioning in demonstrator rats or simply the presence of demonstrator itself. The authors need to explain why such group was not assessed.

  1. In Experiment 2, why did the authors only assess the relationship between “total” 50-kHz calls emitted and observational fear learning? Different subtypes of 50-kHz calls have been categorized [e.g., Behav Ecol Sociobiol 72, 14, https://doi.org/10.1007/s00265-017-2427-9 (2018)], for example, “frequency-modulated 50-kHz calls (step or trill calls, the lower right of Figure 5B)” with high level of frequency modulation and “flat 50-kHz calls (the lower left of Figure 5B)” that often lack frequency modulation; could you also have analyzed and discussed the relationship between such subtypes of 50-kHz calls and observational fear learning?

  1. In the Results section, there was no description about 50-kHz calls in Experiment 2, at least in the proof that I received (typographical error?).

  1. It is of interest and value to discuss the possible information source of observational fear learning instead of ultrasonic vocalizations, for example, visual cues such as facial expression and/or body posture, odors such as alarm pheromone, and so on.

  1. Page 5, Line 26. Duplication of the parenthesis.

Reviewer 2 Report

In this manuscript, Fendt and colleagues analyse the role of trait anxiety and ultrasonic vocalisations on observational fear learning in male rats. I think the manuscript is very well written and would be fully accessible for the readers. The hypotheses are clearly stated and are well supported by the introduction. The experimental design is clear and elegant, and the experiments are correct to test the working hypotheses. The conclusions are supported by the data and the Discussion is clear and easy to follow.

Overall, I think this manuscript is potentially acceptable for publication in Brain Sciences if the authors can address a few concerns.

Major concerns:

  • In the description of Figure 3a data, the authors claim that 22 kHz calls were slightly different between groups, but that the statistical analysis showed no significant effects of group. I think this is confusing. Statistically speaking, if there are no significant differences between groups, then the groups are similar, not different. This issue should be amended to prevent confusing the readership.
  • Experiments presented in Figure 5 should be described in full in the Results section, not in the Discussion section. Also, the authors should clarify how they distinguished which rat was producing the calls during the OFL session.
  • In the Discussion, the authors point that mid anxious rats are somehow more adaptive that their low or high anxious partners. I am not sure I follow the logic since it is not clear that observational learning confers a higher adaptive value. Perhaps the authors would like to expand these concepts so it is clear why this would be the case. I think the counter argument would also be possible. The data may support that learning from others while there was no actual threat may be less adaptive in a natural environment since the threat is no longer present.
  • I am slightly puzzled by the fact that DEM rats show apparently much less 50 kHz calls during FC than NAÏVE rats do during the retention session. I am aware that is not possible to compare these two due to the experimental design, but if 50 kHz calls are emitted when exploring a new environment, then DEM rats should be doing so early during FC. Is there an effect of time on the number of 50 kHz calls for DEM rats during FC? Note that up to the presentation of the first footshock, both sessions are identical, and the behaviour of NAÏVE and DEM rats should be similar up to that point.

Minor concerns:

  • The shock symbol in Figure 1a is not easy to spot. I would suggest making it more salient by using colour or a bigger size.
  • Please include the group sub-group sizes in each figure panel.
  • The characterisation of the 22 and 50 kHz calls is quite different, with several variables being measured for the 22 call, but only number of events for the 50 call. Please include a justification for this discrepancy.
  • Indicate how latency of 22 kHz calls are measured. Is this taken from the start of the session (i.e. since the animal is placed on the experimental box)?
  • Description of Figure 3d is missing from the legend. Besides, authors should clarify in the figure that 22 kHz calls are most likely produced only by DEM rats. This is stated in main text but is not so apparent in the figure.

Reviewer 3 Report

This study investigated how observational fear leaning related to the basal level of trait anxiety and the specific ultrasonic vocalization during the learning process. Overall, the manuscript is well written, the findings are interesting, the methodology is adequately described, and the statistics is appropriate. Addressing the following issues should further improve the rigor, depth, and mechanistic insights of the study.

Major comments:

  1. The DEMs were sent back to home cage with NAIVEs after fear conditioning. Is it possible that the level of trait anxiety could change after Fear Retention Tests for all three groups? Especially NAIVE group.
  2. Why the freezing time was highest in group of medium trait anxiety? Could high level of anxiety influence the learning ability? The authors should discuss more for this topic. 
